

# Effective connectivity differences in motor network during passive movement of paretic and non-paretic ankles in subacute stroke patients

Marianna Nagy[1], Csaba Aranyi[2], Gábor Opposits[2], Tamás Papp[1], Levente Lánczi[1,3], Ervin Berényi[1], Csilla Vér[4], László Csiba[4], Péter Katona[3], Tamás Spisák[5] and Miklós Emri[2]

[1] Faculty of Medicine, Department of Medical Imaging, Division of Radiology and Imaging Science, University of Debrecen, Debrecen, Hajdú-Bihar, Hungary
[2] Faculty of Medicine, Department of Medical Imaging, Division of Nuclear Medicine and Translational Imaging, University of Debrecen, Debrecen, Hajdú-Bihar, Hungary
[3] Department of Diagnostic Radiology, Kenézy University Hospital, Debrecen, Hajdú-Bihar, Hungary
[4] Clinical Center, Department of Neurology, University of Debrecen, Debrecen, Hajdú-Bihar, Hungary
[5] Department of Neurology, University Hospital Essen, Essen, Germany

Corresponding author
Marianna Nagy,
nagy.marianna@med.unideb.hu

## ABSTRACT

**Background:** A better understanding of the neural changes associated with paresis in stroke patients could have important implications for therapeutic approaches. Dynamic Causal Modeling (DCM) for functional magnetic resonance imaging (fMRI) is commonly used for analyzing effective connectivity patterns of brain networks due to its significant property of modeling neural states behind fMRI signals. We applied this technique to analyze the differences between motor networks (MNW) activated by continuous passive movement (CPM) of paretic and non-paretic ankles in subacute stroke patients. This study aimed to identify CPM induced connectivity characteristics of the primary sensory area (S1) and the differences in extrinsic directed connections of the MNW and to explain the hemodynamic differences of brain regions of MNW.
**Methods:** For the network analysis, we used ten stroke patients' task fMRI data collected under CPMs of both ankles. Regions for the MNW, the primary motor cortex (M1), the premotor cortex (PM), the supplementary motor area (SMA) and the S1 were defined in a data-driven way, by independent component analysis. For the network analysis of both CPMs, we compared twelve models organized into two model-families, depending on the S1 connections and input stimulus modeling. Using DCM, we evaluated the extrinsic connectivity strengths and hemodynamic parameters of both stimulations of all patients.
**Results:** After a statistical comparison of the extrinsic connections and their modulations of the "best model", we concluded that three contralateral self-inhibitions (cM1, cS1 and cSMA), one contralateral inter-regional connection (cSMA→cM1), and one interhemispheric connection (cM1→iM1) were significantly different. Our research shows that hemodynamic parameters can be estimated with the Balloon model using DCM but the parameters do not change with stroke.

**Conclusions:** Our results confirm that the DCM-based connectivity analyses combined with Bayesian model selection may be a useful technique for quantifying the alteration or differences in the characteristics of the motor network in subacute stage stroke patients and in determining the degree of MNW changes.

## INTRODUCTION

Numerous studies have used functional magnetic resonance imaging (fMRI) to investigate neural reorganization and functional recovery after stroke (*Calautti et al., 2007*). fMRI univariate analysis of regional activation is valuable for understanding the regional neural substrates associated with cognitive functions (*Ma et al., 2015*). Using active and passive motion-based fMRI experiments (*Lazaridou et al., 2013*; *Cheng et al., 2012*) researchers have typified the post-stroke motor dysfunction and the corresponding potential cerebral reorganization. In our previous therapeutic study, we examined the effects of passive movement on blood oxygen level-dependent (BOLD) responses in both hemispheres (*Vér et al., 2016*). Our results showed that passive movement of the paretic ankle increased BOLD responses in the contralateral pre-and postcentral gyrus, superior temporal gyrus, central opercular cortex, and in the ipsilateral postcentral gyrus, frontal operculum cortex and cerebellum.

Various brain areas are responsible for the execution of movements: the primary motor cortex (M1), supplementary motor area (SMA) and the premotor cortex (PM). The M1 controls the execution of movement by generating neural impulses (*Penfield & Boldrey, 1937*) and the PM regulates the initiation of different motion patterns (*Hoshi & Tanji, 2000*). The role of the SMA is also planning movement, but this is an essential brain area that is activated even when the motion is not performed but only being thought about (*Tanji & Shima, 1994*). These regions create the motor network (*Biswal et al., 1995*) which is primarily responsible for conducting and controlling a wide variety of movements (*Lam et al., 2018*).

In the literature, many researchers used fMRI to investigate the connections and connectivity patterns amongst motor areas during active movement of paretic upper limb after stroke compared to controls. These studies (*Rehme et al., 2011*; *Bajaj et al., 2016*) showed that task demands alter the intensity and volume of brain activity and that these activation changes are specific to certain brain regions.

Dynamic Causal Modeling (DCM) for fMRI is commonly used for analyzing effective connectivity patterns of brain networks. The connection strength between brain regions is assessed at the underlying neuronal level rather than the observed hemodynamic level (*Friston, Harrison & Penny, 2003*). In DCM connectivity is expressed in Hz that shows the rate at which a region's activity is mediated to another region that is, directly connected to DCM provides a fully Bayesian framework to estimate the connectivity strengths of neural interactions between brain regions as well as the regional self-inhibitory effects

and regional hemodynamic parameters. DCM considers several variables such as hemodynamic response, flow induction, activity-dependent signal, change in volume and the level of deoxyhemoglobin. It uses the extended Balloon model (*Friston et al., 2000*) to describe hemodynamic changes due to neuronal activity. For studying the effective connectivity of several diseases DCM has been successfully used (*Seghier et al., 2010*).

Cerebrovascular disorders may affect the shape of the hemodynamic response function (HRF) of BOLD signal. In such patients, HRFs can have lower amplitudes, longer time to peak (TTP) and deeper initial dips (*Altamura et al., 2009*; *Roc et al., 2006*). These observations were also described in cases of retained neuronal activity (*Binkofski & Seitz, 2004*; *Röther et al., 2002*) and in relation to altered cerebral hemodynamics (*Altamura et al., 2009*; *Hamzei et al., 2003*), suggesting that decreased BOLD signals might reflect the reduction of neuronal activation or be the result of neurovascular uncoupling.

Our study investigates whether the impairment of cerebral hemodynamics in subacute stroke patients is related to changes in the BOLD signal HRF.

Task-dependent effective connectivity among motor cortex regions has been documented at rest and during whole-hand fist closing (*Grefkes et al., 2008*; *Nowak et al., 2008*). In able-bodied individuals, the strength and sign of neural coupling between motor areas is modulated by the task (i.e., rest, unilateral, bilateral).

Investigations from the upper limb suggest that the intensity and volume of activity in the primary sensory and motor cortices (S1 and M1) and in the cerebellum are sensitive to movement rate (*Saleh et al., 2016*). In contrast with M1, S1 and the cerebellum, the effect of upper extremity movement rate is less robust in the PM and SMA, respectively (*Inman et al., 2012*). Less is known about the brain activation pathway changes during continuous passive movement (CPM) of a lower extremity. Observations from the upper limb provide a framework for understanding how brain activity changes across different lower limb movement tasks. However, whether upper and lower limb movements are similarly controlled with respect to rate and complexity remains unclear because few studies have examined this issue during tasks of the lower limbs (*Vinehout, Schmit & Schindler-Ivens, 2019*). Differences in the characteristics of arm and leg movements suggest that supraspinal control may also be different (*Mehta et al., 2012*; *Cleland & Schindler-Ivens, 2018*).

Many DCM studies aimed to find the best model for the motor network (*Volz et al., 2015*; *Diekhoff-Krebs et al., 2017*) and in most cases, the best model was the fully connected model. For the upper limb, *Rehme et al. (2011)* investigated the temporal evolution of intra-and interhemispheric connectivity during motor recovery from the acute to the early chronic phase post-stroke. They analyzed 17 possible models and found the fully connected model provided the best results through groups and series. In a similar study, *Bajaj et al. (2016)* studied the connectivity pattern of the motor network in the affected and unaffected hemisphere during finger-tapping task with healthy and paretic arms. From the eight defined connectivity models, they found that the full model fitted the measured data well in the affected hemisphere when the task was completed by the affected hand. In similar research, *Volz et al. (2015)* compared the changes in motor network connectivity during upper and lower limb motion. They examined 39 models and also

found that the best was the fully connected model. In contrast, we defined model families depending on the external stimulus and the connectivity patterns of the S1 region and use the Bayesian model selection technique, which provides evidence for one model over other based upon evidence ratios (i.e., Bayes factors) or differences in log evidence (*Congdon, 2007*).

In our previous publication (*Aranyi et al., 2017*), we examined potential sources of systematic motion artifacts in stroke using fMRI concentrating on those causing stimulus-correlated motion on the individual-level and separated the motion effect change on the fMRI signal from the activation-induced modification at the population level. To allow the models accounting for the sensory dimension of CPM (sensory-feedback), we added the S1 and designed model families to map its connectivity pattern to the motor system.

The aim of the study was to examine the motor network and the connectivity differences in both hemispheres during CPM of non-paretic and paretic ankles in subacute stroke patients in the context of the neuroanatomical and functional background. We specifically focused on the role of the S1 within the network and investigated these modified networks' connectivity differences during CPM of non-paretic and paretic ankles in stroke patients. For the network analysis, we used a DCM-based Bayesian model selection to determine (1) the relationships between the S1 and the motor network and (2) to identify the brain regions that are handling the external stimulations. After the DCM model selection, we (3) investigated the extrinsic connection strength differences between the two CPMs and (4) the dissimilarity of the hemodynamic characteristics of brain regions belonging to the investigated networks.

Note that in contrast to most DCM studies, we leveraged the ability to make inferences about neuronal and hemodynamic coupling. This is particularly prescient in the context of stroke research, where lesions can produce both neuronal and neurovascular disconnections.

## MATERIALS AND METHODS

### Subjects

Ten stroke patients (mean time of stroke: 18.2 days (SD = 11.4); mean age: 64 years (SD = 7.2); female/male distribution: 5/5) were selected from a therapeutic study (*Vér et al., 2016*). Patient data, location of the lesion and National Institutes of Health Stroke Scale (NIHSS) scores are summarized in Table 1. Each patient provided written informed consent prior to participation.

Due to ischemic stroke, patients had moderate or severe lower limb paresis. In our previous study (*Vér et al., 2016*), we applied the NIHSS to assess the functional state and the severity of the stroke in the patients. NIHSS is one of the most commonly used scales in the clinical field; increasing scores indicate less functional states of the patients (*Brott et al., 1989*). Table 2 shows the definitions of mild, moderate and severe paresis.

The inclusion criteria were ischemic stroke confirmed via clinical investigations and computed tomography (CT). We selected subacute stage patients who were less than 30 days post-stroke. Pavlova et al. defined acute stage 1 day to 1 week, subacute stage 1

**Table 1 Demographics, pathology data of stroke patients, clinical characteristics and NIHSS scores.**

| Patient/gender | Age (years) | Lesion type and topography (by CT) | Time of stroke | Severity of lower limb paresis | NIHSS |
|---|---|---|---|---|---|
| 1/Male | 63 | Cerebral infarct in right MCA region | 1 Month | Left sided mild-moderate paresis | 1 |
| 2/Female | 65 | Cerebral infarct in left MCA | 1 Month | Right sided severe paresis | 3 |
| 3/Female | 62 | Bilateral lacunar infarcts, no fresh lesion | 1 Month | Right sided moderate paresis | 3 |
| 4/Male | 56 | Cerebral infarct in right MCA region | 1 Month | Left sided moderate paresis | 3 |
| 5/Male | 52 | No fresh lesion | 9 days | Left sided moderate paresis | 3 |
| 6/Female | 82 | Hypodens lesions in right hemisphere | 6 days | Left sided moderate paresis | 2 |
| 7/Female | 75 | Cerebral infarct in left MCA, old cerebral infarct in right MCA region | 12 days | Right sided mild paresis | 2 |
| 8/Female | 71 | Lesions frontal horns and cella media in right hemisphere | 11 days | Left sided moderate paresis | 2 |
| 9/Male | 59 | No fresh lesion | 5 days | Right sided mild paresis | 1 |
| 10/Male | 58 | No fresh lesion, old lesions in basal ganglia | 9 days | Left sided moderate paresis | 2 |

**Note:**
NIHSS, National Institutes of Health Stroke Scale.

**Table 2 Definitions of mild and moderate paresis and severe paresis (6[th] item of the National Institutes of Health Stroke Scale).**

| Definition of mild and moderate paresis (6[th] item of NIHSS) | 1 point | Drift; leg falls by the end of the 5-second period but does not hit the bed |
|---|---|---|
| | 2 points | Some effort against gravity; leg falls to bed by 5 seconds but has some effort against gravity |
| Definition of severe paresis (6[th] item of NIHSS) | 3 points | No effort against gravity; leg falls to bed immediately |
| | 4 points | No movement |

week to 1 month and chronic stage is more than 1 month after stroke (*Pavlova, Semenov & Guekht, 2019*).

Ability to cooperate during the MRI measurement was an inclusion criterion.

This study was approved by the Regional and Institutional Research Ethics Committee of the Scientific Committee of the University of Debrecen, Clinical Center (DE OEC RKEB/IKEB 3772-2012; DE OEC RKEB/IKEB 3983–2013).

## Functional magnetic resonance imaging

Functional and structural images were acquired at the Kenézy University Hospital, Debrecen, Hungary using a 1.5 Tesla Siemens Magnetom Essenza MR scanner. After the 3D T1-weighted MP-RAGE structural image acquisition (TE = 4.73 ms, TR = 1,540 ms, TI = 800 ms, flip angle = 15 slices with $0.9 \times 0.9 \times 0.9$ mm voxels) two series of functional whole-brain images were obtained for every subject using a BOLD contrast sensitive gradient-echo echo-planar sequence (TE = 42 ms, flip angle = 90, in-plane resolution = $3 \times 3$ mm; volume TR = 4,000 ms, axial slice thickness = 3.3 mm). The two series involved the task of passive movement of the left or the right ankle separately. Both series comprised 100 volumes that lasted 400 s containing 40 s-long blocks of active

and inactive periods. Following an initial resting period, active and inactive blocks alternated throughout the session. During the inactive blocks, no stimulus was applied, whereas in the active blocks, slow (~1Hz) CPM of the left or right ankle was performed by the physiotherapist. The legs and the hip of the patients were fastened to the bed.

## Data analysis

### Image preprocessing

Before preprocessing, the left and the right sides of the structural and functional images of patients with left hemispheric lesion were mirrored. This step facilitated a pooled population-level statistical analysis for all the patients and prevented the need to split the population into two cohorts depending on the side of the stroke.

The image processing pipeline followed the steps used in previous DCM studies regarding motor control in stroke patients (*Grefkes et al., 2008*; *Saleh et al., 2016*). To erase non-brain areas from the functional and structural scans, the brain extraction tool of FSL was used (*Smith, 2002*). The high resolution, brain extracted T1 images were spatially standardized to match the symmetric template of MNI152 space (*Grabner et al., 2006*) using the linear and non-linear registration utilities of the FSL package (*Jenkinson et al., 2012*). We verified the correctness of the image transformation results via visual inspection.

For further analysis, the first three volumes of each functional dataset were excluded to avoid the equilibrium effect of T1 images. The fMRI images were motion-corrected with the MCFLIRT utility of FSL software (*Jenkinson et al., 2002*). The same software was applied to extract the six motion parameters (three rotations and three translation components of rigid body transformations). The fMRI data were then co-registered to the extracted anatomical brain image and spatially transformed to MNI152 space using the deformation field obtained from the T1 standardization process.

On the functional images, we applied an isotropic Gaussian smoothing with 8 mm full width half maximum (FWHM).

### Independent component analysis

After independent component analysis (ICA) (*Jung et al., 2001*), we selected components corresponding to motor network areas (M1, PM and SMA) after visual inspection of the statistical Student-t maps and characteristic time-series of the components on individual level. We completed this search by identifying the primary somatosensory cortex (S1) to examine the effects of the touching of the patients' legs by the physiotherapist during the passive movement task. S1 is responsible for delivering information about the stimulation of the skin, the intestinal mucous membrane of the internal organs, and about the position of the body parts (*Borich et al., 2015*). We chose the highest z value within each cluster as the center of the regions used in DCM analysis.

## Dynamic causal modeling

Dynamic Causal Modeling is used to describe the causative structure of coupled dynamic systems. Using the observable phenomenon of functional connectivity, which can be

measured by correlations, for example, and a special model that describes the observed statistical dependencies, we can map the relationships between different brain areas.

We used a stochastic variant of the DCM algorithm (*Li et al., 2011*) that accommodates endogenous or random fluctuations in hidden neuronal states. This variant is useful in most cases because modeling state-noise is challenging. Allowing for random neuronal fluctuations results in an estimation scheme more robust to model misspecification. A drawback of this method compared to the original deterministic DCM is that the parameter estimation procedure, also referred to as model inversion is much more computationally demanding. To limit the parameters with which connection strength can be estimated, we restricted our DCM model space to combine extrinsic connections and direct stimulating effects on brain regions.

## Systematic building of model-space

For building a DCM model-space for both fMRI sessions, we created bilateral models in relation to the contralateral and ipsilateral brain hemispheres to the moved limb. For finding the most probable connectivity architectures in the motor network based on the measured data, we defined a two-sided base model: the extrinsic (i.e., between-region) directed connectivity between PM, SMA, and M1 regions were fully connected within both hemispheres, and inter-hemispheric connections were set between the M1s and SMAs (Fig. 1A). To investigate the connections between S1 and other regions within both sides of the movement, we defined four possible network extensions: (1) S1 does not connect with PM, SMA and M1; (2) S1 only connects to M1; (3) S1 only connects to PM; and (4) S1 connects to PM and M1 (Fig. 1B, columns). For checking the target regions of the external stimulus at the contralateral side of movement, we considered three functional variations: (1) the stimulus directed to the PM and S1; (2) the stimulus directed to the PM, S1 and M1; and (3) the stimulus directed to the S1 and M1 (Fig. 1B, rows). Combining these possibilities, we established a model space comprising $4 \times 3 = 12$ model variants (Model 1, Model 2, — …, Model 12), which were arranged into a matrix form (Fig. 1B), to emphasize the factorial structure of this model space. The row view of this arrangement represents the first model family set containing three model families ($F^{stim}_1$, $F^{stim}_2$, $F^{stim}_3$) concerning the direct stimulus effect variants. The column view of this matrix shows the second model family set consisting of four families according to the S1 connection arrangements ($F^{S1}_1$, $F^{S1}_2$, $F^{S1}_3$, $F^{S1}_4$).

Figure 1C shows an example of the combination of the base model and Model 4.

Statistical analysis of between-subject (i.e., group) effects was performed under the winning model using (nonparametric) classical inference. The winning model was identified by pooling the evidence for different models over all subjects studied.

## Comparison of model families and model selection

We used a Bayesian model comparison (BMC) (*Penny et al., 2010*) technique on both model family sets to assess the most probable combination of these connectivity parameters based on the fMRI patient. BMC focuses on a model structure that is, created by defining and building a model space. Model space usually means a set of models, where

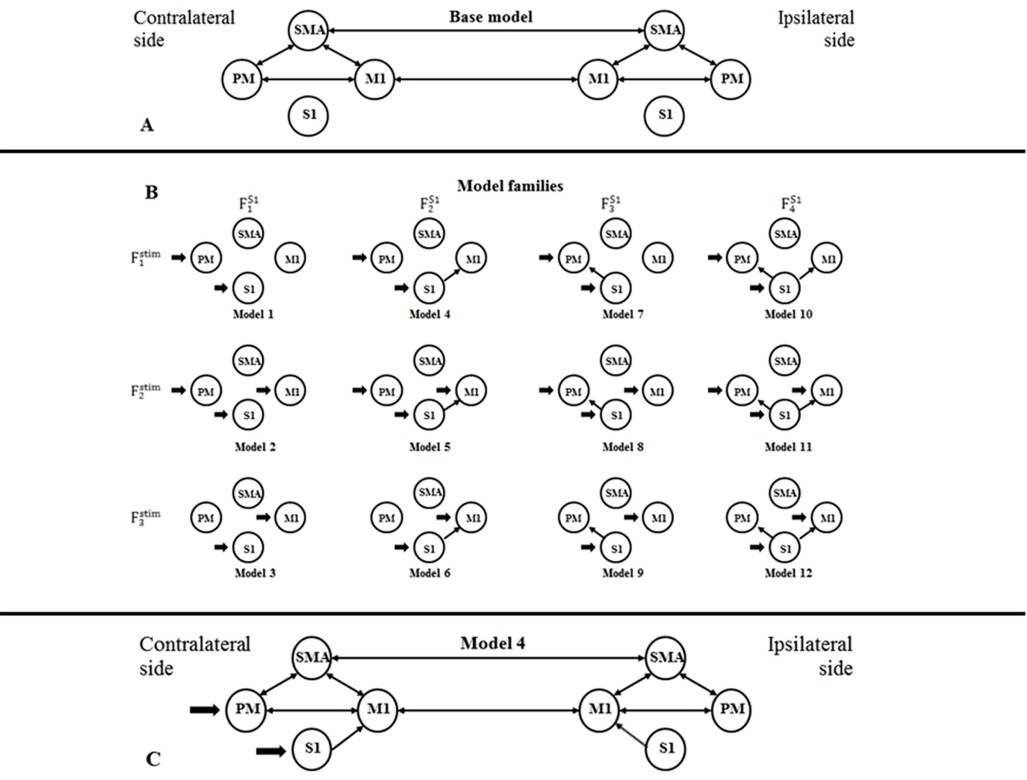

**Figure 1 Model selection for motor network.** (A) Base model: two-way extrinsic (endogenous) connections between PM, SMA, and M1 regions in both hemispheres, and the non-paretic-side M1 and paretic-side M1 areas and non-paretic-side SMA and paretic-side SMA regions were connected. (B) Differences of Model variations organized by two families: The external inputs are shown in the rows, which was only considered on the contralateral side. The columns show the connection system between the S1 and the motor network. (C) An example of the combination of the base model and Model 4.

each model assigns specific endogenous connections. The BMC procedure determines the model that best describes how the data are generated by computing the expected and exceedance probability of each model (*Penny et al., 2004*; *Stephan et al., 2009*). Expected probability shows the probability that a given model generated the measured data. Exceedance probability indicates the probability that a given model is more likely than any other models in the comparison. After the two BMC calculations, we identified the best model family from both family sets, and we considered the mutual model of these selected families as the winning model, which can be described by the measured data.

## Hemodynamic parameters

Neuronal activity leads to fMRI data by a dynamic process characterized by a Balloon model (*Buxton, Wong & Frank, 1998*) and BOLD signal model (*Stephan et al., 2007*) for each brain region. This defines how changes in neuronal activity cause changes in blood oxygenation that are measured with fMRI.

Dynamic Causal Modeling not only investigates the connection system during the model inversion but also estimates the regional hemodynamic parameters of the Balloon

model: hemodynamic signal decay (D), transit time (T) and the ratio of intra-and extra-vascular components of gradient echo-signal (E) (*Stephan et al., 2007*). An increase in the signal decay (D) reduces the regional cerebral blood flow (rCBF) response to any input and suppresses the undershoot. The effect of increasing transit time (T) is to slow down the dynamics of the BOLD signal regarding to the flow changes. The parameter E reflects the efficacy of the following synaptic activity to generate the signal and the potency of the stimulus in obtaining a neuronal response (*Friston et al., 2000*).

In this study, we examined the hemodynamic parameters in subacute stroke patients calculated by the DCM of the winning model for statistical analysis.

### Statistical analysis

Because the non-paretic and paretic ankle CPMs induced lateralized brain activations, we had to relabel the brain regions' names in the point of view of their laterality before statistical analysis. Therefore, we used the ipsi-, and contralateral nomenclature (noted by i and c prefixes, respectively) as Rehme and Saleh (*Rehme et al., 2011*; *Saleh et al., 2016*) used for denoting the position of brain regions relative to the two types of stimuli, that is, cM1, cSMA, cPM, cS1 and iM1, iSMA, iPM, iS1. This allowed us to compare the connectivity strengths and hemodynamic parameters of the activated (or passive) regions during the CPMs of the paretic and the non-paretic ankles.

The expected distributions of the self-inhibition and inter-regional connections are different, because self-connections are log scaling parameters of a negative a priori value ($-0.5$ Hz) to ensure system stability in DCM, while extrinsic connections are not scaled and have a prior expectation of 0 Hz. Thus, we separately investigated the normality of data using the Shapiro–Wilk tests. Because these tests showed non-normality, we performed a mass of Monte-Carlo-based exact permutation tests to statistically characterize the stimulus-related differences between the paretic and non-paretic ankles in the elements of the endogenous connectivity matrix (matrix A) and the parameters of modulatory effects (matrix B). Similarly, neither external stimulus strength (matrix C) nor hemodynamic parameters were found normal distribution, so we applied the same comparison technique for these data.

To correct for multiple comparisons, we applied the false discovery rate (FDR) for adjusting p-values (*Benjamini & Yekutieli, 2001*) calculated by the statistical tests.

## RESULTS

Using BMC selection, we found that during the CPM task, the S1 connection with the M1 and PM ($F^{S1}_4$ family) was the most-likely network topology with 0.784 expected probability and 0.998 exceedance probability (Figs. 2A and 2B). For direct effect model family selection, we observed that the stimulus driving S1, M1 and PM ($F^{stim}_2$ family) was the most probable model with 0.845 expected probability and 0.999 exceedance probability (Figs. 2C and 2D). Using these results, we selected Model11 (Fig. 3) as the winning model for the statistical analysis.

As the Shapiro–Wilk tests resulted in $p < 0.001$ values in all datasets, we used Monte-Carlo-based exact permutation tests in all connectivity and hemodynamic
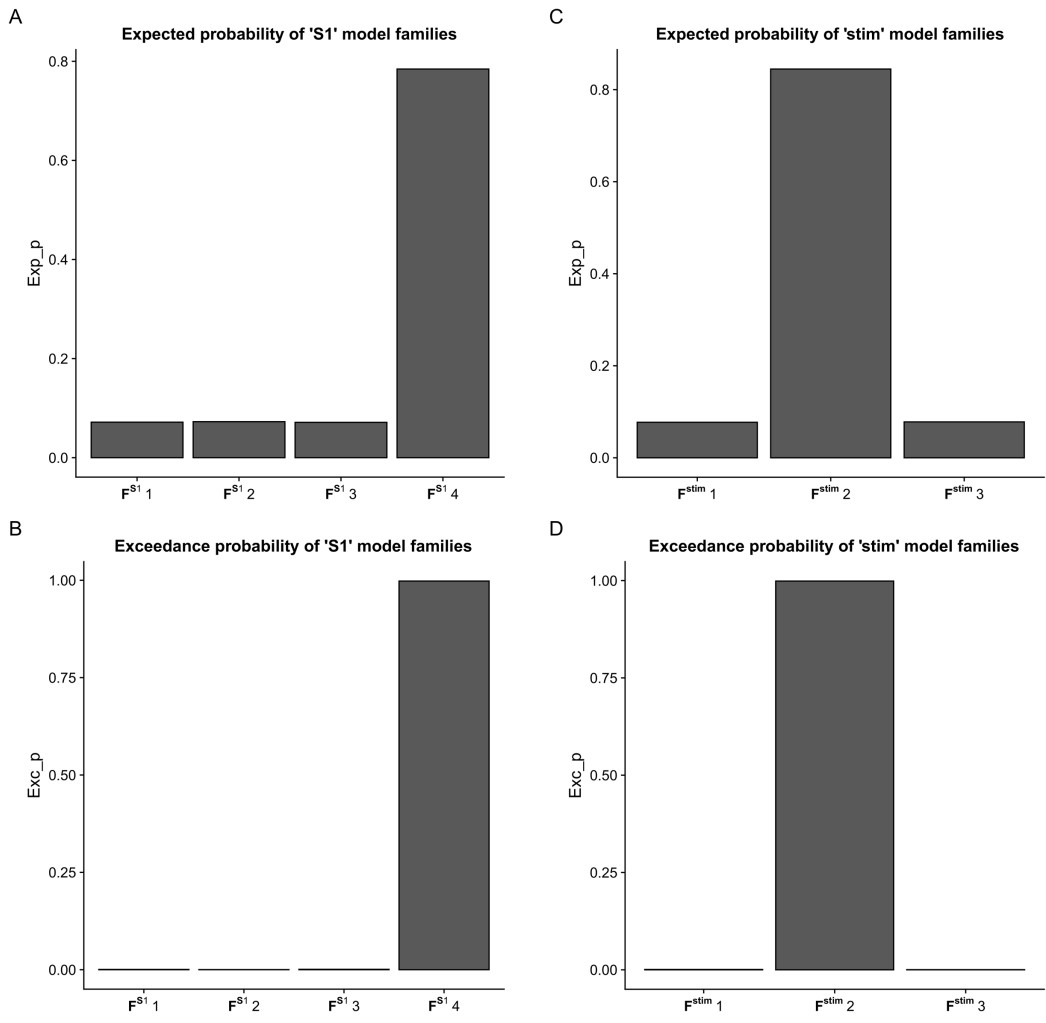

**Figure 2 Results of the Bayesian Model Comparison (BMC) for the F^S1 and F^stim model family sets.**
Results of the Bayesian Model Comparison (BMC) for the $F^{S1}$ and $F^{stim}$ model family sets. Models of both non-paretic and paretic ankle continuous passive movement (CPM) was included in family-wise comparisons. The most probable network topology types were the $F^{S1}_4$ family, with expected probability (Exp_p) of 0.784 and exceedance probability (Exc_p) 0.998 (A and B). For the direct effect model family selection, the $F^{stim}_2$ family was selected with 0.845 Exp_p and 0.999 Exc_p (C and D).

parameter comparisons. Following the statistical analysis of the endogenous connectivity matrix of the winning model (Model 11), we summarized the results in Table 3.

Using the FDR-corrected *p*-values (pFDR), we concluded that three contralateral self-inhibitions (cM1, cS1 and cSMA), one contralateral inter-regional connection (cSMA→cM1), and one interhemispheric connection (cM1→iM1) were significantly different during the comparison of the two CPMs (Fig. 4).

Comparing the mean values, we showed that the paretic CPM caused stronger self-inhibition in cM1 and cS1 and weaker self-inhibition in SMA (Figs. 5A–5C). The connection between cSMA and cM1 dramatically changed (Fig. 5D): during the non-paretic CPM, the cSMA excited the neural activity of cM1, which in turn inhibited the

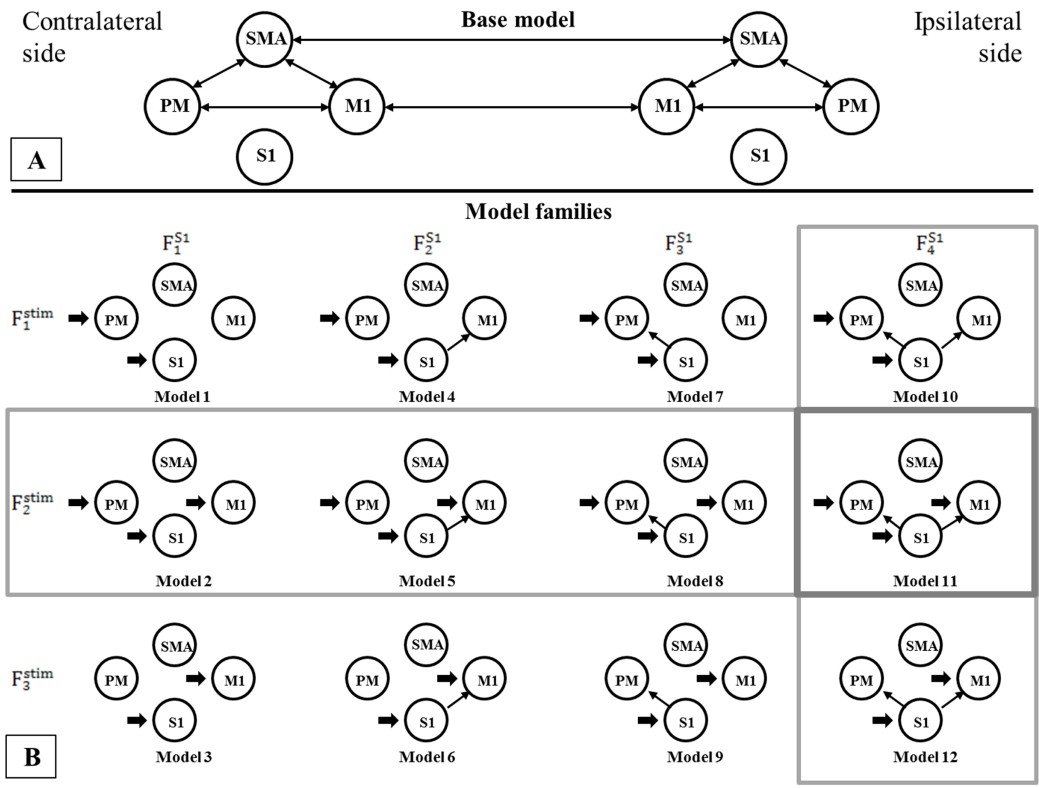

**Figure 3 Comparison of model families.** Comparison of model families. (A) Base model: two-way extrinsic (endogenous) connections between PM, SMA and M1 regions in both hemispheres, and the non-paretic-side M1 and paretic-side M1 areas and non-paretic-side SMA and paretic-side SMA regions were connected. (B) Panel B shows the 12 model variations organized in four $F^{S1}$ families (columns) and three $F^{stim}$ families (rows). Based on BMC results, we found that these models have the most evidence with the highest expected probability that connects S1 to both M1 and PM cortices ($F^{S1}_4$ denoted by— vertical gray rectangle). In the case of the direct effects of the stimulus changing between M1, PM, and S1 areas, the combination of all three regions provides the most probable pattern for stimulating the network for both the non-paretic and paretic sides ($F^{stim}_2$ denoted by—horizontal gray rectangle). Based on this the best model is Model 11, indicated by a gray rectangle.

paretic stimulus. The inter-hemispheric cM1→iM1 connection shown showed stronger excitation during the paretic CPM in relative to the non-paretic ankle's movement (Fig. 5E).

The statistical examination did not show any significant differences neither in the analysis of the external stimulus (matrix C) or in the Balloon model parameter comparisons.

## DISCUSSION

In this work, we examined the connection topology of the S1-extended motor network and investigated the differences in passive-movement related effective connectivity between the paretic and non-paretic limbs during CPM in subacute stroke patients. As the exact effective connectivity structure of the motor network in stroke is not well known, we applied an fMRI-based model-search procedure to identify the model family that best fits the motor network (*Kahan & Foltynie, 2013*; *Penny et al., 2010*). According to previous

**Table 3** Summarized statistical table of extrinsic (endogenous) connection strength analysis of the winning model performed by Monte-Carlo-based exact permutation tests following False Discovery Rate (FDR) correction.

| Connections | Mean connection strength during non-paretic ankle movement | SD of connection strengths during non-paretic ankle movement | Mean connection strength during paretic ankle movement | SD of connection strengths during paretic ankle movement | $p$ value | FDR corrected $p$ value |
|---|---|---|---|---|---|---|
| cM1→cM1 | −0.0925 | 0.0227 | −0.1389 | 0.0489 | 0.0119 | 0.0395 |
| cM1→cPM | 0.0224 | 0.1058 | 0.1086 | 0.0742 | 0.0510 | 0.0756 |
| cM1→cSMA | 0.0903 | 0.1060 | 0.0636 | 0.1118 | 0.5905 | 0.3992 |
| cM1→iM1 | 0.1649 | 0.0438 | 0.2651 | 0.1101 | 0.0127 | 0.0395 |
| cPM→cM1 | 0.0174 | 0.0667 | 0.0648 | 0.0597 | 0.1069 | 0.1073 |
| cPM→cPM | −0.1339 | 0.0295 | −0.1051 | 0.0388 | 0.0778 | 0.0939 |
| cPM→cSMA | 0.1345 | 0.0896 | 0.0592 | 0.0671 | 0.0485 | 0.0744 |
| cS1→cM1 | 0.03680 | 0.0361 | 0.0361 | 0.0514 | 0.9768 | 0.5236 |
| cS1→cPM | −0.0168 | 0.0401 | −0.0064 | 0.0694 | 0.726 | 0.4496 |
| cS1→cS1 | −0.1021 | 0.0281 | −0.1688 | 0.0579 | 0.0045 | 0.0395 |
| cSMA→cM1 | 0.0886 | 0.0511 | 6.3100e-06 | 0.0871 | 0.0123 | 0.0395 |
| cSMA→cPM | 0.244 | 0.1376 | 0.1206 | 0.1464 | 0.0690 | 0.0887 |
| cSMA→cSMA | −0.1780 | 0.0340 | −0.1350 | 0.0336 | 0.0133 | 0.0395 |
| cSMA→iSMA | 0.2755 | 0.0797 | 0.3023 | 0.0896 | 0.4771 | 0.3493 |
| iM1→cM1 | 0.1752 | 0.0834 | 0.2795 | 0.1319 | 0.0488 | 0.0746 |
| iM1→iM1 | −0.1062 | 0.0287 | −0.1303 | 0.0413 | 0.1447 | 0.1400 |
| iM1→iPM | −0.0003 | 0.1588 | 0.0155 | 0.1001 | 0.7919 | 0.4712 |
| iM1→iSMA | 0.0727 | 0.0605 | 0.0932 | 0.0924 | 0.5682 | 0.3900 |
| iPM→iM1 | −0.0024 | 0.1094 | −0.0237 | 0.0260 | 0.5641 | 0.3883 |
| iPM→iPM | −0.1310 | 0.0517 | −0.1332 | 0.0544 | 0.9277 | 0.5107 |
| iPM→iSMA | 0.0304 | 0.0757 | 0.1112 | 0.1245 | 0.0966 | 0.1031 |
| iS1→iM1 | 0.0382 | 0.0849 | 0.0363 | 0.0716 | 0.9615 | 0.5197 |
| iS1→iPM | −0.0414 | 0.1473 | −0.0286 | 0.0712 | 0.8337 | 0.4840 |
| iS1→iS1 | −0.1870 | 0.0784 | −0.1391 | 0.0473 | 0.1042 | 0.1063 |
| iSMA→cSMA | 0.3144 | 0.1231 | 0.2658 | 0.0669 | 0.2867 | 0.2439 |
| iSMA→iM1 | 0.1054 | 0.0548 | 0.0262 | 0.0893 | 0.0274 | 0.0591 |
| iSMA→iPM | 0.1260 | 0.0971 | 0.1808 | 0.1623 | 0.3723 | 0.2952 |
| iSMA→iSMA | −0.1316 | 0.0337 | −0.1709 | 0.0393 | 0.0298 | 0.0615 |

**Note:**

SD, standard deviation; p, probability; c, contralateral; i, ipsilateral; M1, primary motor cortex; PM, premotor cortex; SMA, supplementary motor area; S1, primary somatosensory cortex.

research, in this study, we used the DCM-based effective connectivity technique to describe the motor network properties during the applied CPM stimulations. We chose this method because it helps to understand the causal architecture of the modeled networks by considering the temporal variation in the neural activity estimated by the BOLD signal. We have used a conservative approach to compare extrinsic connectivity between paretic and non-paretic stimulations of the investigated patient group. This nonparametric approach to classical inference at the between-subject level is conservative because it tests for differences in each connection separately (using FDR to adjust for multiple

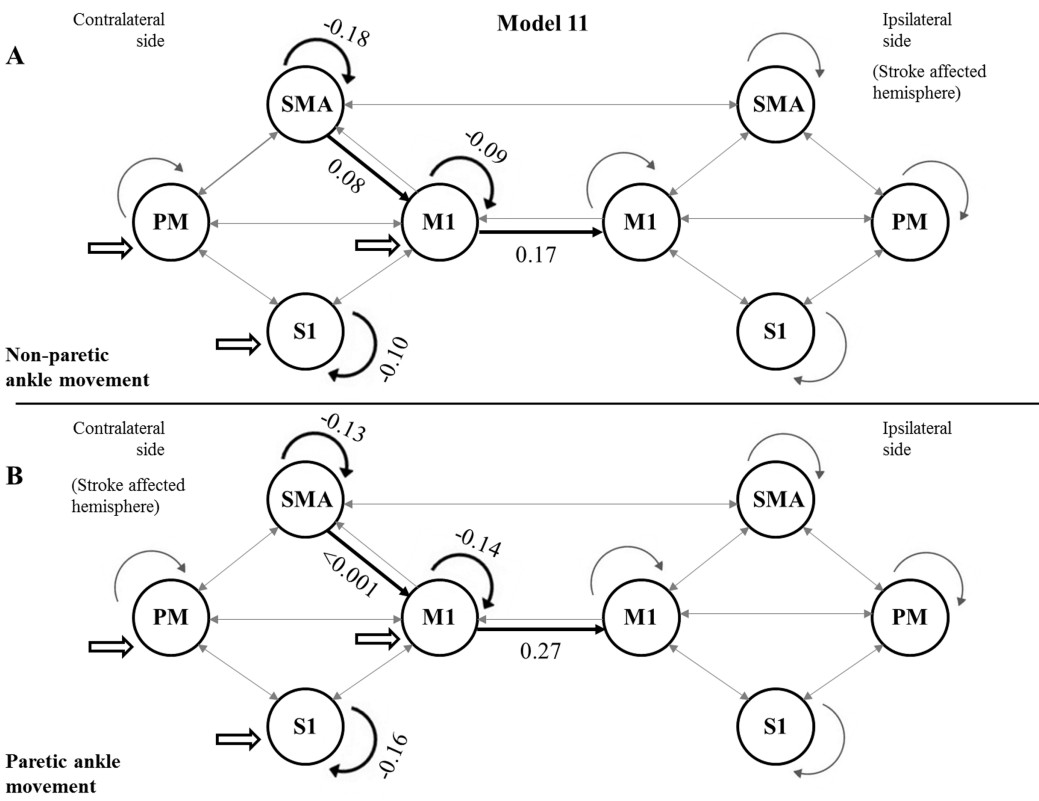

**Figure 4 Mean values of significantly different endogenous connection (matrix A) strengths (see *p*-values in Table 3) of the best model (Model 11).** Bold arrows show the direction of significant connections. (A) Significant differences during non-paretic ankle movement: cSMA→cSMA (mean value = −0.18); cSMA→cM1 (mean value = 0.08); cM1→cM1 (mean value = −0.09); cM1→iM1 (mean value = 0.17) and cS1→cS1 (mean value = −0.10). (B) Significant differences during paretic ankle movement: cSMA→cSMA (mean value = −0.13); cSMA→cM1 (mean value = −0.005); cM1→cM1 (mean value = −0.14); cM1→iM1 (mean value = 0.27) and cS1→cS1 (mean value = −0.16). The empty arrows show the target location of external stimulus. The looping arrows represent the self-inhibitory effects.

comparisons). We could have used a multivariate test (e.g., canonical covariates analysis). This would have been more sensitive and would have highlighted significant stimulation differences. However, we would not have been able to assign a unique FDR's *q* value to each connection. Finally, it should be noted that more recent analyses of between-subject effects on effective connectivity in dynamic causal modeling would normally use parametric empirical Bayes (*Friston et al., 2016*; *Kass & Steffey, 1989*). However, our classical inference is sufficient for our purposes—and represents a simple way of accommodating between-subject random effects.

For model selection, we defined two model families containing four and three models according to the model-combinations of the S1 connections and the external stimuli modeling, respectively. The applied BMC selection showed that the external sensory stimulation bound to the S1 and PM regions, and the S1 had causal connections only with M1 and PM. A model-family-based Bayesian model selection was also applied by *Saleh et al. (2016)*, who examined the interactions between regions that may modulate the

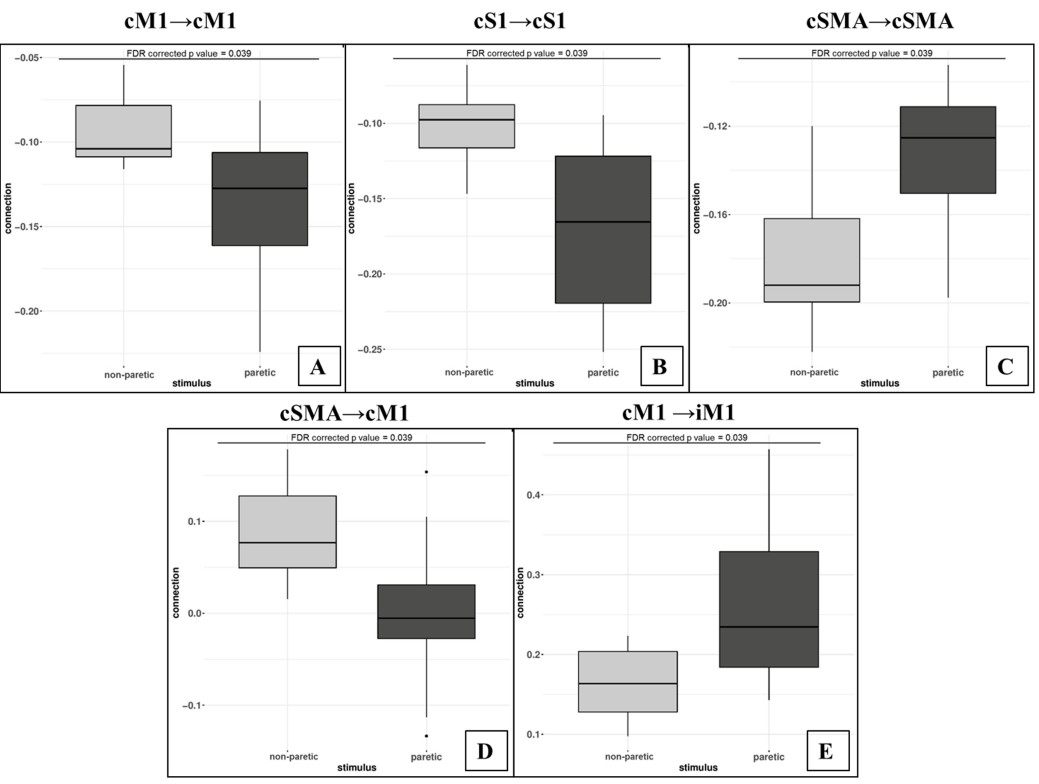

**Figure 5 Comparison of mean values.** On the top of the images can be seen the FDR corrected *p* value differences between paretic and non-paretic ankle movement. (A) The paretic CPM caused stronger self-inhibition in cM1. (B) The paretic CPM caused stronger self-inhibition in cS1. (C) The paretic CPM caused weaker self-inhibition in SMA. (D) The cSMA→cM1 connection changed and, the cSMA excited the neural activity of cM1. (E) The cM1→iM1 connection showed stronger excitation during the paretic CPM.

activation of the ipsilesional motor cortex during visual mirror feedback of unaffected hand movement in stroke patients. They also identified a non-fully connected topological scheme as the best model.

After statistical comparison of the extrinsic connections of the winning model during the non-paretic and paretic CPM, we concluded that three contralateral self-inhibitions (cM1, cS1 and cSMA), one contralateral inter-regional connection (cSMA→cM1), and one interhemispheric connection (cM1→iM1) were significantly different. Some neuroimaging studies reported that during the movement of stroke-affected paretic limbs, a significant neural activity could be observed in the regions of the contralesional hemisphere (*Calautti et al., 2007*; *Ma et al., 2015*; *Lazaridou et al., 2013*; *Badillo, Vincent & Ciuciu, 2013*; *Ward et al., 2003*). *Nowak et al. (2008)* demonstrated that overactivity in contralesional M1 occurs early after stroke, affecting the improvement of these brain regions after the vascular incident. *Grefkes et al. (2008)* also showed that the time after stroke is an essential factor influencing brain motor network analysis.

The human motor network in the hemispheres contains several parts, including the primary motor cortex (M1), SMA and premotor cortex (PM), which territories show ordered representation of the human body, called as the somatotopic representation.

All the parts of the above territories send collateral fibers into the another hemispheres through the corpus callosum, which fibers reaches the same somatotopic territories (homotopic regions) (*Van den Heuvel & Hulshoff Pol, 2010*). However, anatomical connectivity shows difference between the upper and lower extremities, for example, the activation of $M1_{hand}$ inhibit the contralateral $M1_{hand}$ territory, whereas the activation of $M1_{foot}$ accused facilitatory coupling in the contralateral hemisphere (*Volz et al., 2015*). The SMA and PM have strong input into the M1, therefore this adjacent territories can enhance the motor output of M1 pyramidal cells (*Dancause et al., 2005*; *Dum & Strick, 2005*). The primary somatosensory cortex (S1) also able to modify peripheral movements independently (a small part of the corticospinal tract originating from the S1) or dependently from M1 in healthy patients or in stroke survivors. Here the anatomical basis is the U-fibers which caused a strong connectivity between the neighboring gyruses (*Borich et al., 2015*). According to human data the increased peripheral somatosensory inflow helps in the reorganization of M1 after stroke (*Borich et al., 2015*). This connection between the M1 and S1, the literature uses the sensorimotor synchronization (SMS).

The different SMA-M1 connection strength is a common finding in many motor-based DCM studies (*Ward et al., 2003*; *Wang et al., 2016*; *Moulton et al., 2017*). This connectivity, present for simple and complex tasks alike, has been attributed to the respective role of the SMA and M1 in voluntary upper limb movements (*Moulton et al., 2017*). *Pool et al. (2013)* showed that movements at higher frequencies are linked with a linear increase in neural coupling strength, especially from contralateral SMA to contralateral M1. This result shows that SMA cooperates with variations in hand motor performance. Based on this and our results, we concluded that the same alteration occurs in the case of the lower limb. The order of magnitude of the contralateral SMA→M1 connection was different in the two CPMs: the SMA increased the neural activity of M1 by 0.085 Hz (sd: 0.0496 Hz) during the non-paretic side activation, this effect altered to −0.0053 Hz (sd: 0.0845 Hz) in the opposite CPM, which means that this interaction occurs only in the non-paretic case. *Diekhoff-Krebs et al. (2017)* tested the hypothesis that interindividual variability in behavioral responses to excitatory repetitive transcranial magnetic stimulation (rTMS) in stroke patients and healthy control group is related to interindividual differences in the network connections of the stimulated region. Their results revealed that a stronger connection exists between the SMA and the M1 regions before intermittent theta-burst stimulation (iTBS) intervention at the affected arm in patients with better motor performance. Our results support this statement because we also showed a strong relationship between SMA and M1 regions.

The 50% stronger cM1–iM1 interaction (0.2761 Hz vs 0.1705 Hz) during the paretic CPM may indicate a partial adaptive compensation for the injured motor cortex by the non-affected M1 after stroke. Most previous studies examined the functions of the upper limb after stroke. *Grefkes et al. (2008)* showed that the inhibitory influences on movements of the paretic hand from the contralesional to the ipsilesional M1 correspond with the degree of motor disability. They recommended that the motor loss of patients with a single subcortical lesion is connected to pathological interhemispheric interactions

between the main motor regions. This can be explained why we found a difference between interhemispheric connections. The observed connectivity differences suggesting that the differences are region-specific, a residual uncertainty as to localization remains, that is, deep-rooted to cytoarchitectonic probability maps. Functional connectivity studies have been conducted in stroke yet, the changes in tissue composition at the site of the lesion is at various stages of necrosis and gliosis, affecting the BOLD signal (*Frías et al., 2018*). It cannot be excluded that the change in tissue composition might have an effect on functional connectivity. Our results demonstrate that stroke can affect the functional connectivity of regions distant to the infarct, specifically S1, potentially further compromising motor performance.

We showed that the hemodynamical parameters (Balloon model's D, T and E) of the regions of the motor networks (MNW) were statistically similar during the two stimulations. This result suggests that the detected differences in the connection strengths originated from real neural activity, and the hemodynamic change had no confounding effect during the measurements.

Stroke recovery is a complex mechanism that possibly the results of substitution, compensation of functions and combination of restoration (*Hara, 2015*). Many studies addressed the recovery of motor skills after an intervention procedure, and it is well documented that the healthy brain regions take over the functioning of the damaged areas (*Jiang, Xu & Yu, 2013*; *Brown et al., 2009*). In a study on motor recovery following rehabilitation, *Arya et al. (2011)* proposed that the recovery could be compensatory motor recovery or real motor recovery, which occurs when different connections that are unharmed send instructions to the same damaged muscles to perform the motor orders. Neuronal reorganization and plasticity after a stroke takes place during the first 6 months following stroke and involve brain regions distant to the affected area (*Li, 2017*). According to *Zeiler & Krakauer (2013)* after ischemic stroke, both spontaneous and intervention-mediated recovery from impairment is maximal within 1–3 months. Therefore it is difficult to interpret the outcomes of rehabilitative studies in human stroke patients (*Hara, 2015*).

## LIMITATIONS AND CONCLUSION

One limitation of the present study concerns the small sample size. We found significant changes in effective connectivity despite the low number of patients. Furthermore, the difference between left-or right-sided injuries or the anatomical regions of structural damage on effective connectivity could be better investigated in studies with more extended groups.

Another limitation of this study is related to spontaneous plasticity recovery which takes place in the early post-stroke period. Physiotherapy strategies used during the recovery process affect spontaneous neuroplasticity. According to further studies, the optimal timing to begin rehabilitation after stroke is still not known and it is highly possible that early intervening to impacts cortical reorganization in a beneficial way (*Coleman et al., 2017*; *Cassidy & Cramer, 2017*).

In summary, our results confirm that the DCM-based connectivity analyses combined with Bayesian model selection may be a useful technique for quantifying the alteration or differences in the characteristics of the motor network in subacute stage stroke patients and in determining the degree of MNW changes. However, in stroke, the number of patients who can be involved in these types of fMRI studies is a severely limiting factor, yet.

### Funding
This research was supported by the National Brain Research Program (2017-1.2.1-NKP-2017-00002). The funders had no role in study design, data collection and analysis, decision to publish, or preparation of the manuscript.

### Grant Disclosures
The following grant information was disclosed by the authors:
National Brain Research Program: 2017-1.2.1-NKP-2017-00002.

### Competing Interests
The authors declare that they have no competing interests.

### Author Contributions
- Marianna Nagy analyzed the data, prepared figures and/or tables, authored or reviewed drafts of the paper, and approved the final draft.
- Csaba Aranyi analyzed the data, prepared figures and/or tables, and approved the final draft.
- Gábor Opposits analyzed the data, authored or reviewed drafts of the paper, and approved the final draft.
- Tamás Papp analyzed the data, authored or reviewed drafts of the paper, and approved the final draft.
- Levente Lánczi analyzed the data, authored or reviewed drafts of the paper, and approved the final draft.
- Ervin Berényi conceived and designed the experiments, authored or reviewed drafts of the paper, and approved the final draft.
- Csilla Vér conceived and designed the experiments, performed the experiments, authored or reviewed drafts of the paper, and approved the final draft.
- László Csiba conceived and designed the experiments, performed the experiments, authored or reviewed drafts of the paper, and approved the final draft.
- Péter Katona conceived and designed the experiments, performed the experiments, authored or reviewed drafts of the paper, and approved the final draft.
- Tamás Spisák analyzed the data, authored or reviewed drafts of the paper, and approved the final draft.
- Miklós Emri analyzed the data, prepared figures and/or tables, and approved the final draft.

## Human Ethics

The following information was supplied relating to ethical approvals (i.e., approving body and any reference numbers):

This study was approved by the Regional and Institutional Research Ethics Committee of the Scientific Committee of the University of Debrecen, Clinical Center (DE OEC RKEB / IKEB 3772-2012; DE OEC RKEB / IKEB 3983–2013).

## Data Availability

Data is available at the University of Debrecen Electronic Archive: https://dea.lib.unideb.hu/dea/handle/2437/277097.

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
