# Peer review of "Effective connectivity differences in motor network during passive movement of paretic and non-paretic ankles in subacute stroke patients"

_PeerJ, doi:10.7717/peerj.8942_

## Round 0.1 · original submission · Major Revisions

Dear Authors,

We have received 3 detailed reviews and we thank the reviewers for their time (note that the comments of Reviewer 2 are predominantly in their attached PDF).

Please revise the manuscript according to the important comments given by the peer reviewers.

·

Basic reporting

Please see General reporting

Experimental design

Please see General reporting

Validity of the findings

Please see General reporting

Additional comments

Comments to editors

I thought that this was an excellent submission. It applies dynamic causal modelling to the sensorimotor system of people with paretic and nonparetic stroke. There were a couple of technical issues in their application of dynamic causal modelling that need to be addressed. However, I think that these can be resolved with a minor revision – and that they will not change the conclusions of this study. I have tried to suggest how the authors could rationalise their analysis in my comments below.

I hope that these comments help you in your evaluation.

Comments to authors

I enjoyed reading this concise and compelling report of a dynamic causal modelling study of subacute stroke. In many respects, I thought that this report was exemplary. Your motivation for analysing effective (directed) connectivity in this clinical setting was clear. I also thought that the description of your analyses was excellent. Having said this, there are some technical issues that you need to resolve. I suspect that this will require a reanalysis of your data; however, it should not take too long. Perhaps you could consider the following:

Major points

The main problem with the current analyses is that you have included modulatory effects (parameterised by the B matrices) in your DCM. This is not necessary and introduces a redundant parameterisation. The reason that you do not need the modulatory effects is that there is only one experimental factor in your design (namely, the active versus inactive block conditions). You only need to use modulatory effects in multifactorial designs. Put simply, this means that there is only one thing perturbing your distributed neuronal responses and therefore you only need to estimate the endogenous connectivity (the A parameters) for each subject. This will not make any difference to your results (note that you go got no significant differences between the B matrices).

The second analytic issue is the fact that you did your model comparisons within each (paretic and nonparetic) group separately. When testing for differences in connectivity between subjects, it is important not to do anything that is specific to each group before you do the final comparison. This means that you should combine all of your subjects (from both groups) in the same (random effects) family wise model comparison. Looking your results, you will get exactly the same family as you have currently used.

Finally, although the original description of DCM used the label "intrinsic" to refer to endogenous connectivity, it is now used to refer to within-region connectivity (i.e., self-connections). Can you replace "intrinsic" with "extrinsic" throughout the manuscript? When you first mention extrinsic connectivity introduce it with: "… extrinsic (i.e., between-region) directed connectivity"

In summary, these changes should simplify your analysis. First, simply omit any reference to modulatory effects in the introduction and results (and tables). Second, you only need to describe one set of family wise Bayesian model comparisons (for both groups combined).

In the discussion, I think you should include a qualification of how you analysed your between subject effects. I would recommend adding:

"In this work, we have used a conservative approach to compare extrinsic connectivity between the clinical groups. This nonparametric approach to classical inference at the between-subject level is conservative because it tests for differences in each connection separately (using false discovery rate to adjust for multiple comparisons). We could have used a multivariate test (e.g., canonical covariates analysis). This would have been more sensitive and would have highlighted significant group differences. However, we would not have been able to assign a unique q value to each connection. Finally, it should be noted that more recent analyses of between-subject effects on effective connectivity in dynamic causal modelling would normally use parametric empirical Bayes (Friston et al 2016, Kass & Steffey 1989). However, our classical inference is sufficient for our purposes – and represents a simple way of accommodating between-subject random effects."

Minor points

Abstract: please say: "in extrinsic connections of the MNW and to explain the haemodynamic…"

Replace "collected by" with "collected under"

Replace "and the stimulus modelling" with and input stimulus modelling"

I would also add the following to the abstract or introduction:

"Note that in contrast to most DCM studies, we leveraged the ability to make inferences about neuronal and haemodynamic coupling. This is particularly prescient in the context of stroke research, where lesions can produce both neuronal and neurovascular disconnections."

Line 61: please delete "and cortical activation"

Line 122: change "during the upper" to "during upper"

Line 126: I would say: "which provides evidence for one model over other based upon evidence ratios (i.e., Bayes factors) or differences in log evidence."

Line 133: change "top" to "to"

Line 139: delete "algorithm"

Line 216: replace "interconnected or subdivided" with "coupled"

Line 232: replace "connection system" with "connectivity architectures"

Line 242: say: "a model space comprising 4x3 = 12 model variants.… which were arranged into a matrix form, to emphasise the factorial structure of this model space.

Revise analysis of the modulation effect paragraph. Please remove any reference to modulatory effects.

Line 258: I would replace "further statistical…" with "Statistical analysis of between-subject (i.e., group) effects was performed under the winning model using (nonparametric) classical inference. The winning model was identified by pooling the evidence for different models over all subjects studied."

Line 266: remove "that are modulated by experimental variations"

Line 268: please revise your description of family wise model comparison (please see above)

Line 306: please replace "stimulus binding" with "stimulus driving"

I hope that these comments help should any revision be required.

Reviewer 2 ·

Basic reporting

The main goal of the present study is interesting. However, there is insufficient background provided. The methodology section lacks clarification and there is missing information (see pdf attached for more information). Finally, the reading of the paper is tough and not easy to follow (see my comments in the pdf document).

Experimental design

The methods presented in the study are not described with sufficient detail (see pdf document for more information).

Validity of the findings

Conclusions fall beyond the findings presented in the manuscript. Authors should try to discuss their results from a biological point of view instead of looking for validations in other studies in which the methodology conducted is enough different to draw firm conclusions (see pdf document for further details). Besides, limitations should be addressed: a small sample and the election of the brain regions of interest...

Annotated reviews are not available for download in order to protect the identity of reviewers who chose to remain anonymous.

Reviewer 3 ·

Basic reporting

'no comment'

Experimental design

'no comment'

Validity of the findings

'no comment'

Additional comments

The paper describes some novel findings regarding directed functional connectivity in patients with subacute stroke using a passive movement task.

Major concerns:

- small sample size (n = 10). The patient cohort is pretty inhomogeneous (NIHSS ranges from 1-3). Did the authors consider the paresis severity in their DCM computations as well as in the “standard” GLM fMRI analyses?
- Clear hypotheses are missing
- some patients are probable able to move their ankles, as the NIHSS is only 1. Why did the authors involve a physiotherapist? I guess it is to control the frequency of movements. If so, how was the frequency of the CPM controlled. Would the authors expect a different (GLM or DCM-wise) result, for self-controlled ankle movements?
- Using S1 as DCM region makes only sense in case of touch-related tasks, as done in the present study (i.e. a physiotherapist is moving the ankle). However, this is not a realistic scenario, in case a patient is undergoing normal physiotherapy training. Thus, the observed BMC results (S1 -> M1 and S1 -> PM), is most likely only present in case of external sensory input (e.g. externally triggered CPM). Alternatively, is the selected S1 region corresponding to the feet region in the brain?
- What is an expected probability value of 0.433 and exceedance probability of 0.63 (Figure 2a/b) telling us? Is this a strong or a weak effect (what are the limits, can we relate that to an effect size)?
- DCM still suffers (except of newer versions such as generalized DCM) from the fact that a ton of models can be defined by “knowing” somehow the winner model a priori. How did the authors limit the likelihood of overfitting (as the authors tested 12 models)?

Abstract:
- The word “lateralized” needs clarification
- the term “signal decay” needs a short introduction
- The conclusion is speculative. To what extend is a DCM-based result informs us anything about any appropriate rehabilitation procedure?

Introduction:
- first sentence (line 50): The terms “directly” and “indirectly” are confusing. For example, are they related to functional or anatomical connections?
- line 61/62: In which brain regions?
- line 78: This section needs to be slightly extended, so that the reader can understand what is modeled on the neuronal level by DCM.
- line 93: this needs more explanation. Why can we exclude a cerebral hemodynamic impairment?

Figure 4: The last sentence is difficult to understand, i.e. what is meant by “the location of external stimulus”? In addition, the twisted arrows are not explained. Please provide the p-value of the results.

Methods:
- line 270: What is the difference between the winning or best model? In this paragraph, authors need to explain the measures D, T, and E in more detail.
- line 274: What is a model inversion?
- line 290: what is the assumption behind the idea that self-inhibition = negative values. What is meant with self-inhibition?

Conclusion:
The first part of the conclusion rather belongs to the Discussion section. Critically, either the results do inform about “during” or “after rehabilitation”, as this is cross-sectional study. Please specify. The last sentence is a limitation. A limitation section should be added.

Discussion:
Lines 346-363 are somewhat redundant as the authors repeat results. Please shorten this part.

---

## Round 0.2 · accepted · Accept

Dear Authors, Your manuscript has been accepted after revision for publication in Peer J.Congratulations!

·

Basic reporting

Please see below

Experimental design

Please see below

Validity of the findings

Please see below

Additional comments

Many thanks for attending to my previous comments – and congratulations on interesting study.